# Precision Medicine in Glaucoma: Artificial Intelligence, Biomarkers, Genetics and Redox State

**DOI:** 10.3390/ijms24032814

**Published:** 2023-02-01

**Authors:** Antonio Maria Fea, Federico Ricardi, Cristina Novarese, Francesca Cimorosi, Veronica Vallino, Giacomo Boscia

**Affiliations:** Department of Ophthalmology, University of Turin, Via Cherasco 23, 10126 Turin, Italy

**Keywords:** biomarkers, artificial intelligence, genetics, glaucoma, ophthalmology, redox state, visual fields, optical coherence tomography, exosomes, precision medicine

## Abstract

Glaucoma is a multifactorial neurodegenerative illness requiring early diagnosis and strict monitoring of the disease progression. Current exams for diagnosis and prognosis are based on clinical examination, intraocular pressure (IOP) measurements, visual field tests, and optical coherence tomography (OCT). In this scenario, there is a critical unmet demand for glaucoma-related biomarkers to enhance clinical testing for early diagnosis and tracking of the disease’s development. The introduction of validated biomarkers would allow for prompt intervention in the clinic to help with prognosis prediction and treatment response monitoring. This review aims to report the latest acquisitions on biomarkers in glaucoma, from imaging analysis to genetics and metabolic markers.

## 1. Introduction

The progressive loss of retinal ganglion cells (RGC) that characterizes glaucoma, the most prevalent cause of permanent blindness in the world, is thought to be the result of a multifactorial neurodegenerative illness. A combination of clinical examination, intraocular pressure (IOP) measurements, visual fields, and structural imaging parameters are currently needed to diagnose glaucoma, classify its severity, and monitor the progression and the response to medical or surgical intervention. A diagnosis may be missed or delayed when visual symptoms are minimal or absent in the early stages of glaucoma.

Clinical, functional, and structural tests now suffer from the necessity for baseline testing and the lengthy lag time needed to determine disease progression and treatment response. Additionally, RGCs may experience malfunction before cell death, which may be reversed with treatment [1]. Thus, there is a critical unmet demand for glaucoma-related biomarkers to enhance clinical testing for early diagnosis and tracking of the disease’s development. The United States National Institutes of Health Biomarkers Definitions Working Group defined a biomarker as “a characteristic that is objectively measured and evaluated as an indicator of normal biological processes, pathogenic processes, or pharmacologic responses to a therapeutic intervention” [2]. Pulse rate and blood pressure are examples of biomarkers, as well as simple chemistry tests and more advanced laboratory or imaging studies. The development of biomarkers for therapeutic utility involves statistical validation of repeatability, specificity, sensitivity, and the evaluation of relevance [3]. Relevance is the capacity of a biomarker to deliver information that will influence clinical decisions and endpoints.

The introduction of validated biomarkers would allow for prompt intervention in the clinic to help with prognosis prediction and treatment response monitoring, as well as to conduct early-phase clinical trials more quickly, for example, to test potential medicines [4]. This review aims to report the latest acquisitions on biomarkers in glaucoma, from imaging analysis to genetics and metabolic markers.

## 2. Method of Literature Search

We searched Pubmed, Web of Science, ScienceDirect, and EMBASE for articles relating to glaucoma and precision medicine. We included articles from 1992 to August 2022 by using the following terms in various combinations: “precision medicine”, “biomarkers”, “artificial intelligence”, “genetics”, “glaucoma”, and “redox state”. Studies were limited to the English language. Relevant articles from the reference lists of the identified articles were manually searched for additional inclusions. However, articles written in other languages with an English abstract were also considered if the abstract provided adequate information. Articles without an English abstract were excluded.

## 3. Artificial Intelligence in Glaucoma

The use of artificial intelligence is expanding rapidly. Machine learning (ML) and deep learning (DL) allowed a more sophisticated and self-programming way to use machines in automatic data analysis. More in detail, in machine learning, a system can automatically improve its performance and learn by itself with experience without being specifically programmed to do so. Specifically, using a convolutional neural network (CNN) architecture, the deep learning algorithm can take in an input image, assign importance (learnable weights and biases) to various aspects/objects in the image, and be able to differentiate one from the other [5]. Similar to neurons derived from the mammalian visual cortex, the neural network’s architecture consists of many hidden layers, each with its specific receptive field and connection to a further layer (Figure 1).

The deep learning network works as a two-step process. The first is the feature learning step, in which convolution, pooling, and activation functions make the ‘jump ahead’ between hidden layers. Secondly, the classification function converts the probability value to a label, providing a clinical output such as healthy or pathologic [6,7].

Although this architecture traditionally provided a high degree of computational power, in recent years, more advanced network architectures have been developed, allowing the system to analyze more complex data sources. AlexNet (2012) was introduced to improve the results of the ImageNet challenge. VGGNet (2014) was introduced to reduce the number of parameters in the CNN layers and improve the training time. ResNet (2015) architecture makes use of shortcut connections to solve the vanishing gradient problem (which is encountered when during the iteration of training, each of the neural network’s weights receives an update proportional to the partial derivative of the error function with respect to the current weight) [8]. The basic building block of ResNet is a residual block that is repeated throughout the network. There are multiple versions of ResNet architectures, each with a different number of layers. Inception (2014) increases the network space from which the best network is to be chosen via training. Each inception module can capture salient features at different levels [9].

Traditional metrics assessing the DL algorithm’s quality are sensitivity, specificity, precision, accuracy, positive predictive value, negative predictive value, and area under the receiver operating curve (AUC).

It is known that the early detection of glaucoma could eventually preserve vision in affected people. However, due to its clinical history of being symptomatic only in advanced stages and when most of the RGCs are already compromised, it is crucial to introduce a tool to detect glaucoma in clinical practice in pre-symptomatic form automatically. Furthermore, it could be of clinical relevance also to find new ways to provide targeted treatment and forecast the clinical progression.

In this scenario, we reviewed the most recent advances in artificial intelligence for either the detection or prediction of glaucoma progression, focusing on various sources of clinical data such as fundus photography (FP), optical coherence tomography (OCT), and standard automatic perimetry (SAP).

### 3.1. Fundus Photography

In clinical practice, ophthalmologists suspect glaucoma by analyzing optic nerve head (ONH) anatomy, cup-to-disc ratio (CDR), optic nerve head notching or vertical elongation, retinal nerve fiber layer (RNFL) thinning, presence of disc hemorrhages, nasal shifting of vessels, or the presence of parapapillary atrophy. However, the diagnostic process could be challenging considering the extreme variance of these parameters [10]. It has been shown that agreement among experts on detecting glaucoma from optic nerve anatomy is barely moderate [11]. Furthermore, with standard fundus photography, not only the variability of anatomy could be misleading, but also the parameters of acquisition such as exposition, focus, depth of focus, contrast, quality, magnification, and state of mydriasis.

In this scenario, artificial intelligence algorithms can extract various optic disc features and automatically detect glaucoma from fundus photographs. For example, Ting et al. [7] collected 197,085 images and trained an artificial intelligence algorithm to automatically determine the cup-disc ratio (CDR) with an AUC of the receiver operating characteristic (ROC) curve of 0.942 and sensibility and specificity, respectively, of 0.964 and 0.872. Similarly, Li et al. [12] developed an algorithm based on 48,116 fundus images reporting high sensitivity (95.6%), specificity (92.0%), and AUC (0.986). Although the importance of automatically detecting the excavation of the optic nerve head, it is known that high inter-subject variability characterizes CDR; some large optic nerve heads have bigger cupping even without any sign of glaucoma. To reduce the rate of false positives, other researchers trained a deep learning algorithm to determine the presence of glaucoma based on fundus photographs and implemented it with the visual field severity [13].

Li and coworkers used a pre-trained CNN called ResNet101 and implemented it with raw clinical data in the last connected layer of the network; interestingly, there were no statistically significant changes in AUC, but they found an improvement in the overall sensitivity and specificity of the model, confirming the importance of multi-source data to improve the discriminative capacity of the glaucomatous optic disc [14].

More recently, Hemelings et al. utilized a pre-trained CNN structure relying on active and transfer learning to develop an algorithm with an AUC of 0.995. They also introduced the possibility for clinicians to use heatmaps and occlusion tests to understand better the predominant areas from which the algorithm based its predictions; it is an exciting way of trying to overcome some problems related to the well-known ‘black-box’ effect [15].

The majority of the publications that were analyzed suggested that an automated system for diagnosing glaucoma could be developed (Table 1). The severity of the disease and its high incidence rates support the studies that have been conducted. Deep learning and other recent computational methods have proven to be promising fundus imaging technologies. Some recent technologies, such as data augmentation and transfer learning, have been used as an alternative way to optimize and reduce network training, even though such techniques necessitate a large database and high computational costs.

### 3.2. Optical Coherence Tomography

Optical coherence tomography (OCT) is an essential tool to capture not only the glaucomatous optic disc in two dimensions (2D) but to provide a three-dimensional (3D) visualization, including the deeper structures. It is a technique based on the optical backscattering of biological structures; it has been widely adopted to assess glaucoma damage both on the anterior segment (e.g., with anterior segment OCT to detect angle closure) and posterior segment (e.g., with traditional OCT to detect ONH morphology and RFNL thickness) [50].

For this reason, depending on the input data, it is possible to differentiate five subgroups of deep learning models: (1) models for prediction of OCT measurements from fundus photography, (2) models based on traditionally segmented OCT acquisitions, (3) models for glaucoma classification based on segmentation-free B-scans, (4) models for glaucoma classification based on segmentation-free 3D volumetric data and (5) models based on anterior segment OCT acquisitions.

Thompson et al. showed that it is possible to predict the Bruch membrane opening-based minimum rim width (BMO-MRW) using optic disc photographs with high accuracy (AUC was 0.945) [51]. Similarly, other researchers reported a high AUC for their model to predict RNFL thickness from fundus images [52,53,54]. Asaoka et al. developed a CNN algorithm to diagnose glaucoma based on thickness segmentations of RNFL and ganglion cells with inner plexiform layer (GCIPL) [42,55]. Wang et al. used 746,400 segmentation-free B-scans from 2669 glaucomatous eyes to automatically develop a model to detect glaucoma with an AUC of 0.979 [56].

Maetschke et al. [57] developed a DL model with an AUC of 0.94 using raw unsegmented 3D volumetric optic disc scans. Similarly, Ran et al. [58] validated a 3D DL model based on 6921 OCT optic disc volumetric scans; the AUC was 0.969, with a comparable performance between the model and glaucoma experts. Russakoff et al. used OCT macular cube scans to train a model to classify referable from non-referable glaucoma; despite the quality of the model, it did not perform as expected on external datasets [59].

At last, DL models based on AS-OCT have been developed to detect the presence of primary angle closure glaucoma (PACG), such as the one proposed by Fu et al. [60]. Xu et al. further developed this type of algorithm to predict the PACG as well as the spectrum of primary angle-closure diseases (PACD) (e.g., primary angle-closure suspect, primary angle-closure) [61].

The papers cited clearly demonstrated that using DL on OCT for glaucoma assessment is effective, precise, and encouraging (Table 2). Despite that, prior to implementing DL on OCT monitoring, more research is required to address some current challenges, including annotation standardization, the AI “black box” explanation problem, and the cost-effective analysis after integrating DL in a real clinical scenario.

### 3.3. Standard Automatic Perimetry

Visual field testing represents a fundamental exam for diagnosing and monitoring glaucoma. In distinction from the fundus photographs and OCT, it allows the interpretation of the functionality of the whole visual pathway. Given the importance of visual function testing for the detection and clinical forecast of glaucoma, many researchers recently developed DL algorithms using the complex quantitative data it contains. Asaoka et al. [69] trained a DL algorithm to automatically detect glaucomatous visual field loss with an AUC of 0.926; the performance of their model was higher if compared to other machine learning classifiers methods, such as random forests (AUC 0.790) and support vector machines (AUC 0.712).

Elze et al. [70] employed archetypal analysis technology to obtain a quantitative measurement of the impact of the archetypes or prototypical patterns constituting visual field alterations. Similarly, Wang et al. developed an artificial intelligence approach to detect visual field progression based on spatial pattern analysis [71].

Given the importance of predicting visual loss patterns in glaucoma patients, specifically for prescribing a personalized treatment, researchers have developed interesting tools to predict the probability of disease progression based on visual field data. DeRoos et al. [72] were able to compare forecasted changes in mean deviation (MD) on perimetry at different target pressures using a machine-learning technique called Kalman Filtering (KF). KF is a machine-learning technique derived from the aero-spatial industry that compares the course of the disease of a single patient to a population of patients with the same chronic disease; in this scenario, it could potentially predict the rate of conversion to glaucoma in patients with ocular hypertension as well as disease progression in the future for patients with manifest glaucoma (Table 3). [73,74].

The possibility of low-cost screening tests for the disease has been made possible by the consistent demonstration of deep learning models’ ability to detect and quantify glaucomatous damage using standard automated perimetry automatic assessment. Additionally, it has been demonstrated that DL enhances the evaluation of the damage on unprocessed visual field data, which could enhance the utility of these tests in clinical practice. As already stated, the validation of new diagnostic tests, despite how exciting AI technologies may be, should be based on a rigorous methodology, with special attention to the way the reference standards are classified and the clinical settings in which the tests will be adopted.

## 4. Biomarkers and Precision Medicine in Glaucoma

Following the definition given by the National Institutes of Health, a biomarker is “a characteristic that is objectively measured and evaluated as an indicator of normal biological processes, pathogenic processes, or pharmacological responses to a therapeutic intervention”. Notably, a molecular biomarker may be helpful for clinicians to identify the early phase of glaucoma to guarantee the most accurate and precise approach. To display this function, biomarkers recognized as valid for clinical practice must have reasonable specificity, sensitivity, and reproducibility. By now, several researchers have focused their attention on the identification of new molecular markers characteristic for glaucoma manifestation and progression. Among these, multiple metabolites and proteins have been investigated in different biological fluids—such as tear film, vitreous, aqueous, serum, and plasma—as they are potentially involved in molecular and cellular pathways causing the damage of neuronal cells in people suffering from glaucoma [79]. Given the considerable number of molecules identified in recent years, our purpose is to provide an overview of the main categories of the biological markers suitable for improving glaucoma detection.

### 4.1. Proteins and Hormones as Biomarkers in Glaucoma

Proteins are large molecules composed of one or more chains of amino acids displaying several biological functions. The dysregulation of the levels of the same proteins has been reported to be associated with glaucoma [80]. For instance, Farkas et al. reported that upregulation in the retina of iron-regulating proteins ceruloplasmin, ferritin, and transferrin and their mRNA was correlated to glaucoma [81]. These results suggested the involvement of iron, copper, and antioxidant mechanisms in the pathogenesis of glaucoma [81]. In the same way, Lin and coworkers revealed an association between higher ferritin levels and glaucoma in a South Korean population [82]. Wang and colleagues compared the levels of matricellular proteins in the aqueous humor from acute primary angle closure (APAC) and non-glaucomatous cataract patients, finding a significant upregulation in the first group [83]. Another research from González-Iglesias et al. evaluated, through quantitative immunoassay, the alteration of 17 overexpressed proteins in the intact serum of patients with primary open-angle glaucoma (POAG) and pseudo-exfoliation (PEX) glaucoma compared to the healthy control group [84]. They suggested that these candidate markers were part of a system linked to inflammatory and immune mechanisms [84].

Another interesting field of research considers the activity of growth factors and cytokines and their role in glaucoma development. Several studies showed the involvement of brain-derived neurotrophic factor and nerve growth factor, quantified in serum and tears, in regulating retinal ganglion cell survival. These results suggest that altered levels of these markers are associated with different stages of glaucoma [85,86,87]. Gupta and colleagues evaluated whether the tear film cytokines can represent a biomarker in the early POAG [88]. Their results showed that the tear film values of the majority of cytokines were significantly lower in the group with glaucoma and, consequently, may be tested as a marker of early POAG [88]. Another research from Guo and coworkers investigated secreted frizzled-related protein-1 (SFRP1) and transforming growth factor-β2 (TGFβ2) levels in the aqueous humor of patients suffering from several types of glaucoma [89]. They reported an increase in TGFβ2 and a decrease in SFRP1 levels in patients with POAG. Conversely, both TGFβ2 and SFRP1 were correlated to a higher IOP in patients with acute angle-closure glaucoma [89]. Other authors suggested a possible involvement of hormones in glaucoma development due to estrogen receptors in retinal ganglion cells [80]. Li et al. evaluated the correlation of 17-b-estradiol and interleukin-8 with primary angle closure glaucoma in postmenopausal women [90]. They found an increase in interleukin-8 and a decrease in 17-b-estradiol in the serum correlated with a visual field (VF) progression [90].

Moreover, other studies suggested that estrogens have a protective role through the downregulation of cytokine formation, causing the blockage of one of the possible pathways of ganglion cell loss [91,92]. Canizales et al. explored the expression of superoxide dismutase 1 (SOD1) in the aqueous humor of patients with POAG [93]. They found a significant decrease in SOD1 mRNA expression in the POAG group than in the control group [93]. In a subsequent paper, Mirzaei et al. evaluated proteins of the retina and vitreous of controls and glaucomatous eyes, revealing a defect of oxidative phosphorylation and the overexpression of proteins correlated to the classical complement pathway, and previously reported to be associated with glaucoma (Table 4) [94].

### 4.2. Metabolic Biomarkers in Glaucoma

The research of metabolic products in glaucomatous patients may represent, for operators, a possibility to implement current diagnostic tools. Among metabolites, one of the most investigated is homocysteine (Hcy). Hcy results from methionine metabolism implicated in oxidative stress mechanisms and vascular dysregulation. Consequently, increased levels suggest a correlation between glaucoma and the vascular system [96]. Research from Lee and coworkers reported a correlation between plasmatic Hcy and glaucomatous retinal nerve fiber layer (RNFL) defect in a group of 78,049 South Korean people [96]. A subsequent study confirmed these results by evaluating the plasma levels of Hcy, L-cysteine (Cys), and hydrogen sulfide (H_2_S) in people with POAG, OHT, and normal tension glaucoma (NTG) and normal individuals [97]. They found reduced levels of H2S and increased levels of Hcy and Cys in the glaucoma group, particularly in POAG [97]. In the same way, López-Riquelme et al. explored plasmatic values of Hcy, endothelin-1 (ET-1), vitamins A, B12, E, and folic acid in three populations: patients with POAG, NTG, and control group [98]. While vitamin E was lower in NTG, Hcy and ET-1 were increased in POAG, suggesting a correlation with endothelial dysfunctions. Conversely, vitamin E lowering may result from oxidative processes involved in glaucoma pathogenesis [98]. In contrast with these results, a paper published in 2016 by Leibovitzh and colleagues, including 11,850 Israelian patients, reported that Hcy levels and IOP did not show any association [99].

Baumane et al. investigated the role of the atrial natriuretic peptide (ANP) system in glaucoma pathogenesis by quantifying the N-terminal fragment of the pro atrial natriuretic peptide (NT-proANP) levels in the aqueous humor and the plasma of people with glaucoma and cataract [100]. Their results highlighted an increase in NT-proANP in patients with POAG, supporting the theory of possible involvement of ANP in the pathogenesis of glaucoma [100]. Javadiyan et al. explored the serum level of symmetric dimethylarginine (SDMA) and asymmetric dimethylarginine (ADMA) in people with advanced glaucoma compared to the control group [101]. These two metabolites interfere with the nitric oxide (NO) formation system, and NO is an antioxidant protecting and supporting endothelial cell activity. Both SDMA and ADMA showed an increase in the serum of patients with advanced glaucoma, indicating a possible role of the NO pathway in the development of glaucoma [101].

Another interesting and frequently investigated marker is uric acid (UA). UA is the result of human purine metabolism and displays an antioxidant function [102]. Despite the fact that we have several reports in the literature debating on the association with POAG, the pathogenetic effect of UA is not clarified. Moreover, there is no consensus on the direction of the UA effect. These results might be due to differences in ethnicity or to the presence of other confounding factors. One of the possible explanations is linked to the antioxidant activity of UA [103]. Li et al. evaluated UA levels in the serum of patients with POAG compared to controls, finding a significantly higher concentration in the latter group [104]. Thus, their conclusions further support the theory of oxidative stress action in glaucoma pathogenesis.

A different field of research investigated biomarkers potentially modified by the action of anti-glaucoma medications [105]. For instance, Kotikoski and coworkers showed that NO metabolites changes might be disguised by glaucoma medications [106]. In the same way, other papers revealed that the number of inflammatory cells in the conjunctiva of patients undergoing long-term therapy for glaucoma was remarkably increased with a consequent overexpression of inflammatory markers (Table 5) [107].

### 4.3. Antibodies as Biomarkers in Glaucoma

To date, multiple studies indicate the presence of an autoimmune component in glaucoma development. We have several reports in the literature describing autoantibodies—such as anti-HSP60, anti-a-A-crystalline, a-B-crystalline, HSP27 in NTG, and anti-HSP70, phosphatidylserine, glycosaminoglycans, g-enolase, α-fodrin, vimentin, glutathione-S-transferase, retinaldehyde binding protein, glial fibrillary acidic protein, retinal S-antigen, and neuron-specific enolase—and all possibly correlate with glaucoma [108,109]. For instance, Grus and coworkers, in 2006, showed that people with glaucoma have characteristic differences in serum antibodies from the controls [110]. Furthermore, Western blot results reported an increased level and an improved reactivity to α-fodrin, an autoantibody previously described in other neurodegenerative diseases [110]. Joachim and colleagues analyzed the IgG autoantibody patterns against antigens of the retina, the optic nerve head, and the optic nerve in the serum of a glaucomatous population compared to a control group [111]. The POAG group presented a higher reactivity against retinal antigens, while in NTG, the autoimmune activity was addressed against the optic nerve head [111]. The following research compared the IgG antibody profile against retinal antigens in the aqueous humor of patients with POAG, pseudoexfoliation glaucoma (PEX), and healthy individuals [112]. While no alteration was noted between POAG and PEX, both groups showed significant variations if compared to controls [112]. Tezel et al. conducted an immunoproteomic comparison of serum IgG on 111 people with POAG and 49 controls, revealing several potential biomarkers of glaucoma [113]. Subsequently, Schmelter et al., evaluating IgG in the sera of 13 POAG patients, discovered 75 peptides corresponding to variable IgG domains, presenting substantial glaucoma-related changes [114].

More recently, Beutgen et al. used serological proteomics analysis to examine autoantibody profile and revealed three new biomarkers, PGAM1, CALD1, and VDAC2, that, merged with the previously discovered markers HSPD1 and VIM, can differentiate glaucomatous from controls with a specificity of 93% and a sensitivity of 81% [108].

Hohenstein-Blaul and coworkers, using an experimental autoimmune glaucoma animal model, showed a loss of retinal ganglion cells, independent from IOP elevation, correlated to antibody depositions and increased values of microglia [109]. They revealed an association between neuronal damage and changes in autoantibody reactivity. These results suggest a potential role of autoantibody profiling as a biomarker for glaucoma (Table 6) [109]. 

### 4.4. Exosomes in Glaucoma

Extracellular vesicles are a group of lipidic bilayered structures that include microvesicles, apoptotic bodies, and exosomes. Among them, with a mean size of 50 nm, the latter is the smallest subtype [115]. They were found to be secreted by different types of cells, displaying several roles, such as the mediation of intercellular communication via receptor–ligand mechanisms and the transport of their load to other sites [115]. Recently the interest of researchers in the role of exosomes in ocular diseases is emerging. Lerner et al. explored the activity of exosomes originating from the non-pigmented ciliary epithelium (NPCE) on the Wnt signaling pathway, a central regulator of TM, revealing that these vesicles influence such pathways [116]. Subsequent research from the same group analyzed NPCE exosomes, finding 182 proteins and 584 miRNAs displaying several functions, such as intercellular adhesion and the deposition of extracellular matrix in TM [117]. Stamer et al. showed that TM exosomes contain proteins causing glaucoma, such as myocilin, and, consequently, hypothesized a possible function on the dysregulation of IOP [118].

Moreover, some authors proposed extracellular vesicles and exosomes as therapeutic approaches for glaucoma-related damages. Mead and coworkers showed that bone marrow-derived stem cell (BMSC) small extracellular vesicles, with an intravitreal administration, have a neuroprotective action for 12 months in a rat model of OHT [119]. In the same way, Pan and colleagues demonstrated that umbilical cord mesenchymal stem cells-derived exosomes implemented retinal ganglion cell maintenance in a mouse model [120].

Since glaucoma is a progressive and irreversible cause of vision loss, an early diagnosis is a crucial aspect in the management of such a disease. A helpful strategy for glaucoma detection includes the research of metabolites and proteins and, consequently, represents a possibility to shed light on unknown molecular pathways implicated in the early stages of the disease. Indeed, in addition to ethnicity, age, and sex, other potential risk factors are represented by the dysregulation of systemic and ocular vascular pathways, the presence of oxidative stress, and altered antibody profiles. One of the main examples is the evaluation of the plasmatic levels of Hcy and Cys in patients with POAG and OHT. Indeed, such results indicate a potential implication of methionine metabolism, related to oxidative stress mechanisms and vascular dysregulation, in the pathogenesis of POAG [96]. Despite that, it would be advisable to consider such markers with caution due to the heterogeneity of disease presentation and the differences in the genetic pattern of individuals. In the same way, other findings frequently reported in the serum and in the tear film of glaucomatous are autoantibodies against ocular tissues. Indeed, following the last reports, we are able, through the antibody profile, to distinguish the serum of glaucomatous from controls with a specificity of 93% [108]. However, it is still unclear whether these alterations in antibody profile are a cause or a consequence of glaucoma-related alterations. Consequently, the evaluation of biological markers detectable with non-invasive tests, such as a blood sample, seems more like an opportunity to obtain additional clinical information on the disease characterization and progression than a real possibility to address the therapeutical approach (Table 7).

## 5. Genetics and Precision Medicine in Glaucoma

The newest and more affordable DNA genotyping allows for the easier identification of the genes involved in the susceptibility for POAG [121,122]. Indeed, for patients with a new diagnosis of POAG, a proper genetic analysis may provide more accurate information on the prognosis and the adequacy of the therapeutic approach. Moreover, in clusters of families affected by hereditary glaucoma, genetic examinations allow closer monitoring of relatives at risk of developing an overt POAG [122]. Furthermore, one of the main advantages of discovering new “genetic markers” is the possibility to identify previously unrecognized biological pathways, causing the disease manifestation and progression, and consequently personalize the treatment of glaucoma.

One of the main reports of gene-supported precision medicine is the predictive gene testing for myocilin-associated glaucoma [123]. Indeed, MYOC gene mutation is associated with POAG, steroid-induced glaucoma, and juvenile-onset glaucoma [123,124,125]. MYOC gene encodes for myocilin, a 55–57 kDa glycoprotein strongly expressed in TM cells. Although the precise mechanism of IOP increase is not thoroughly described, the most accepted theory points to the accumulation of mutated forms of myocilin in the endoplasmic reticulum of the trabecular meshwork, activating a stress response (unfolded protein response) and altering the resistance of this tissue. Mutated forms of myocilin accumulate in the trabecular meshwork (TM) and modify the resistance to aqueous outflow [124].

One of the most extensive research investigating the ability of a genotype to predict the conversion from ocular hypertension (OHT) to POAG was the Ocular Hypertension Treatment Study (OHTS) [126]. This research aimed to assess, in a group of non-Hispanic Whites patients genotyped for variants previously correlated to POAG, the association between the variants and the conversion from OHT to POAG. The authors showed that a single-nucleotide polymorphism (SNP) in TMCO1 is strongly associated with POAG development [126]. TMCO1 has been reported to be a gene strongly implicated with IOP magnitude. Indeed, people with double-risk alleles showed a three-fold increased possibility of converting OHT in POAG compared to those with no risk alleles. Notably, no statistical association was found in the African American subgroup.

Another exciting field of research is the possibility of predicting the progression of the disease in patients already in treatment for POAG. A study by Trikha and coworkers investigated whether genetic loci causing POAG were associated with VF progression [127]. Their paper included 469 Singaporean Chinese patients with five or more reliable VF. Their results showed that only an SNP in TGFBR3-CD27 region was correlated to an increased risk of VF progression (*p* = 0.002; odds ratio, 6.71 per risk allele) [127].

Within the field of precision medicine, pharmacogenomics, which is defined as the branch of genetics concerned with how an individual’s genetic attributes affect the likely response to therapeutic drugs, offers clinicians an excellent opportunity to further personalize the medical approach to the patient. Indeed, there is well-reported evidence that genetic analysis may predict the efficacy of therapy and some of its potential side effects [128]. For instance, McCarty et al. found that an SNP in ADRB2 was associated with an increased IOP-lowering response to topical beta-blockers [129]. ADRB2 is an adrenergic receptor gene, and its mutation results in an agonist-promoted downregulation of such receptor. They reported that the polymorphism rs1042714 in ADRB2 was significantly more prone to manifest an IOP decrease of 20% or more. Similarly, Sakurai et al. investigated the patients’ response to latanoprost [130]. They reported an association between SNPs of the prostaglandin F2α receptor gene and the magnitude of the IOP reduction both in OHT and POAG patients [130]. Indeed, they suggest a correlation between the polymorphism rs12093097 and a lowered response of these receptors to latanoprost. Low et al. reported a case series of five patients with concomitant malignant glaucoma (MG) and genetically confirmed BEST1 gene mutation [131]. A multi-disciplinary approach concluded that a traditional surgical procedure ended up with a poor outcome in four patients and used a different approach with the fifth patient who did not experience MG and had no glaucoma progression five years after a pars plana vitrectomy and the insertion of a pars plana Baerveldt tube, thus providing the proof-of-principle that genetic analysis can be used to select the most appropriate surgical therapy in selected cases [131].

Multiple studies focused on gene polymorphism causing steroid-induced ocular hypertension (SIOH) and glaucoma. In this case, the genes so far involved are GPR158 and HCG22 [132,133,134]. The first one encodes a member of the G protein-coupled receptor family that increases in the trabecular meshwork (TM) cells with glucocorticoid administration. This mechanism improves the barrier function of TM cells monolayer and, consequently, the resistance to aqueous outflow [134]. The second one codifies a mucin protein expressed in TM, which increases after glucocorticoid assumption [132]. These findings have the potential to predict the predisposition to IOP increase in specific individuals before the onset of long-term steroid therapies [133].

Moreover, several authors investigated the role the IL-6 gene displays in regulating fibrosis after filtration surgery and in the progression of POAG [126,135,136,137]. Indeed, Yu-Wai-Man et al. showed that the upregulation of IL-6 and the downregulation of the PRG4 gene contributed to conjunctival fibrosis after glaucoma surgery [135]. Following this research, another paper from Fernando et al. suggested administering targeted siRNA nanocomplexes to transport regulators of fibroblasts to prevent fibrosis after filtration surgery [138]. Nevertheless, it is still not recommended to choose medical and surgical approaches based on genetic testing [122,139].

If pharmacogenomics helps predict the individual’s response to therapeutic drugs, epigenetics, on the other hand, may modify genetic imprinting. Several authors focused on studying the environmental action on gene expression and evaluating different mechanisms to modify DNA regulation. An example was the research from McDonnel and coworkers [140]. They found that hypoxia displays a role in regulating the expression of the pro-fibrotic TGFβ1 and the anti-fibrotic RASAL1 and the whole DNA methylation in glaucomatous trabecular meshwork [140]. A previous randomized controlled trial, the Collaborative Initial Glaucoma Treatment Study, reported the action of smoking on IOP [74]. Notably, IOP was higher in smokers than in non-smokers after nine years in the surgically treated group, while no significant difference was noted in the medically treated group [74]. Their results suggest a possible function of tobacco in altering specific molecular pathways.

As stated above, in recent years, the interest in the field of genetics related to glaucoma has significantly increased. One of the researchers’ main targets is to treat the disease in the earlier phase with a more accurate approach led by genetics. The future of genetic testing aims to offer prognostic information potentially addressing the follow-up strategy and the intensity of the care. One example is the ability of TMCO1 to predict the conversion of OHT to POAG. Despite that, none of the genetic markers reported in the literature seems to be a reliable track to lead glaucoma management due to the relative smallness of the samples and the absence of replicability of such studies [122]. Thus, we are looking forward to larger and multicentric trials, resulting from global cooperation, able to direct operators’ decisions and offer our patients genetic-based precision medicine (Table 8).

## 6. NAD+/NADH Redox State and Glaucoma

To date, the gold standard in the treatment of the POAG aims to reduce the IOP. Despite that, a large group of patients progresses independently from the IOP lowering, indicating the presence of multiple risk factors in glaucoma development and worsening [142]. Among them, a dysregulation of biochemical pathways, such as a dysfunction of mitochondrial activity, may be linked to an increased weakness of the optic nerve head (ONH). Indeed, an alteration of mitochondrial activity, which was previously correlated to other neurodegenerative diseases, causes a lack of oxygenation of the retinal ganglion cells, resulting in damage to the ONH [142]. As stated above, several authors have reported oxidative damage associated with POAG. Notably, the mitochondrial electron chain transfers electrons from the reduced form of nicotinamide adenine dinucleotide (NADH, reduced form of NAD+) to oxygen. Reduced levels of NAD+ represent the “primum movens” for impaired mitochondrial activity and, consequently, for neurodegenerative damages [142]. Thus, the NAD+/NADH-redox state is an essential parameter in the energy production pathway and can potentially be used as a systemic biomarker of susceptibility to POAG [142]. Research from Kouassi and coworkers on 34 POAG individuals showed that nicotinamide, the primary precursor of NAD+, was lowered in the plasma of glaucomatous patients [143]. These results suggest a possible role of nicotinamide supplementation as therapy for POAG [143].

In the same way, Williams and colleagues revealed that oral nicotinamide or genetic therapy, such as a driver of Nmnat1, an enzyme-producing NAD+, was effective as a prevention and treatment for POAG [144]. Similarly, Hui et al. showed that nicotinamide supplementation improved inner retinal survival in glaucoma [145]. Moreover, a nicotinamide supplementation showed to be a suitable inhibitor of poly (ADP–ribose) polymerase 1 (PARP-1), a member of the PARP group of enzymes involved in genome stability maintenance [144,146]. A condition with increased oxidative stress, such as in glaucoma, activates PARP-1, reducing NAD+ levels; thus, its blockage seems a good option for a therapeutical approach to reduce NAD+ depletion [144].

Other researchers focused on the possibility of applying NAD+ producing enzymes as protecting factors. The two main categories are nicotinamide mononucleotide adenylyl transferases (NMNATs) and nicotinamide phosphoribosyl transferase (NAMPT). For instance, Williams and colleagues revealed in a rat model that a genetic therapy, such as a driver of Nmnat1, an enzyme of the NMNATs family, was protective against POAG [144]. Avery et al. observed that Nmnat1 relocalizing proteins that move the latter outside the nucleus improve its neuroprotective function [147]. Indeed, the nuclear Nmnat localization seems to be useless in exerting a neuroprotective activity [147].

Another interesting research from Braidy and coworkers reported that increased activity of inducible (iNOS) and neuronal nitric oxide synthase (nNOS) was linked to an implemented NAD+depletion and cytotoxicity [148]. Consequently, nNOS and iNOS activity blockage effectively inhibits these mechanisms [148]. Since the NOS function was found to be implemented in POAG, reduced nitric oxide may be beneficial for glaucoma-suffering patients [149]. Conversely, the isoform NOS-3, present in vascular endothelial cells and astrocytes, may be neuroprotective, exerting vasodilatation and enhancing blood flow [149].

Another field of research includes sirtuins, a group of enzymes that operate ADP ribosylation and deacetylation [142]. Since sirtuins activity is regulated by NAD+ and nicotinamide concentration and has a neuroprotective function on retinal ganglion cells, their dysregulation may be considered part of glaucoma pathogenesis [142,150]. One example has been reported by Balaiya and coworkers [142]. They tested in vitro the SIRT1 action in sustaining RGC viability by inducing hypoxia with a cobalt chloride (CoCl_2_) administration. Cell apoptosis was evaluated by measuring stress-induced protein kinases activity. They found that higher concentrations of CoCl_2_ increased SIRT1 levels significantly and reduced RGC viability. Moreover, by inhibiting SIRT1 action, the viability of RGC was further decreased (Table 9) [142]. 

## 7. Conclusions

Even though clinically accepted biomarkers exist for diagnosing and treating glaucoma, the demand for novel biomarkers with increased sensitivity and specificity remains. This is especially crucial in the case of glaucoma, as the disease’s effects can be drastically decreased with early detection and appropriate management.

Molecular biology, genome sequencing technologies, and pharmacogenomics are radically changing the development of novel medications in many medical specialties [151]. These developments will likely result in the rapid growth of new proposed therapeutics to delay or even reverse glaucoma-related neuronal atrophy. However, surrogate endpoints may be valid alternatives when getting the actual endpoints would make the study impractical. However, these substitutes must be appropriately evaluated before broad usage in clinical practice. Validation requires evaluating biological plausibility and predictive value and determining how much of the treatment’s effect on clinically significant outcomes can be represented by the surrogate. Such validation studies have not yet been undertaken for any of the putative biomarkers now available for glaucoma, and the scientific community should make efforts to design and conduct such research. The potential for improved precision in diagnosis and treatment is significant in the field of glaucoma.

## Figures and Tables

**Figure 1 ijms-24-02814-f001:**
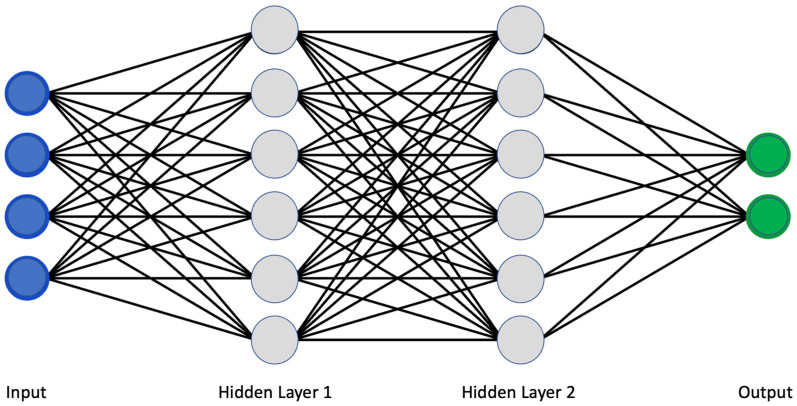
Classical scheme of a convolutional neural network.

**Table 1 ijms-24-02814-t001:** Summary of studies on glaucoma detection using fundus photography.

Author	Year	N. of Images	Structure	SEN	SPEC	ACC	AUC
Kolar et al. [16]	2008	30	FD			93.80%	
Nayak et al. [17]	2009	61	Morphological	100%	80%	90%	
Bock et al. [18]	2010	575	Glaucoma Risk Index	73%	85%	80%	
Acharya et al. [19]	2011	60	SVM			91%	
Dua et al. [20]	2012	60	DWT			93.3%	
Mookiah et al. [21]	2012	60	DWT, HOS	86.7%	93.3%	93.3%	
Noronha et al. [22]	2014	272	Higher order cumulant features	100%	92%	92.6%	
Acharya et al. [23]	2015	510	Gabor transform	89.7%	96.2%	93.1%	
Isaac et al. [24]	2015	67	Cropped input image after segmentation	100%	90%	94.1%	
Raja et al. [25]	2015	158	Hybrid PSO	97.5%	98.3%	98.2%	
Singh et al. [26]	2016	63	Wavelet feature extraction	100%	90.9%	94.7%	
Acharya et al. [27]	2017	702	kNN (K = 2) Glaucoma Risk index	96.2%	93.7%	95.7%	
Maheshwari et al. [28]	2017	488	Variational mode decomposition	93.6%	95.9%	94.7%	
Raghavendra et al. [29]	2017	1000	RT, MCT, GIST	97.80%	95.8%	97%	
Ting et al. [7]	2017	494,661	VGGNet	96.4%	87.2%		0.942
Kausu et al. [30]	2018	86	Wavelet feature extraction, Morphological	98%	97.1%	97.7%	
Koh et al. [31]	2018	2220	Pyramid histogram of visual words and Fisher vector	96.73%	96.9%	96.7%	
Soltani et al. [32]	2018	104	Randomized Hough transform	97.8%	94.8%	96.1%	
Li et al. [12]	2018	48,116	Inception-v3	95.6%	92%	92%	0.986
Fu et al. [33]	2018	8109	Disc-aware ensemble network (DENet)	85%	84%	84%	0.918
Raghavendra et al. [29]	2018	1426	Eighteen-layer CNN	98%	98.30%	98%	
Christopher et al. [34]	2018	14,822	VGG6, Inception-v3, ResNet50	84–92%	83–93%		0.91–0.97
Chai et al. [35]	2018	2000	MB-NN	92.33%	90.9%	91.5%	
Ahn et al. [36]	2018	1542	Inception-v3 Custom 3-layer CNN			84.5% 87.9%	0.93 0.94
Shibata et al. [37]	2018	3132	ResNet-18				0.965
Mohamed et al. [38]	2019	166	Simple Linear Iterative Clustering (SLIC)	97.6%	92.3%	98.6%	
Bajwa et al. [39]	2019	780	R-CNN	71.2%			0.874
Liu et al. [40]	2019	241,032	ResNet (local validation)	96.2%	97.7%		0.996
Al-Aswad et al. [41]	2019	110	ResNet-50	83.7%	88.2%		0.926
Asaoka et al. [42]	2019	3132	ResNet-34				0.965
			ResNet-34 without augmentation				0.905
			VGGI I				0.955
			VGGI 6				0.964
			Inception-v3				0.957
Kim et al. [43]	2019	1903	Inception-V4	92%	98%	93%	0.99
Orlando et al. [44]	2019	1200	Refuge Data Set	85%	97.6%		0.982
Phene et al. [45]	2019	86,618	Inception-v3	80%	90.2%		0.945
Rogers et al. [46]	2019	94	ResNet-50	80.9%	86.2%	83.7%	0.871
Thompson et al. [47]	2019	9282	ResNet-34				0.945
Hemelings et al. [15]	2020	8433	ResNet-50	99%	93%		0.996
Zhao et al. [48]	2020	421	MFPPNet				0.90
Li et al. [49]	2020	26,585	ResNet101	96%	93%	94.1%	0.992

FD = fractal dimensions; SVM = support vector machine; DWT = discrete wavelet transform; HOS = higher-order spectra; PSO = particle swarm optimization; kNN = k-nearest neighbor; RT = Radon transformation; MCT = modified census transformation; NN = neural network.

**Table 2 ijms-24-02814-t002:** Summary of studies on glaucoma detection using OCT technology.

	Author	Year	Outcome Measures	Arch	SEN	SPEC	ACC	AUC
**OCT Fundus**	Thompson et al. [47]	2019	1. Global BMO-MRW prediction	ResNet34				0.945
			2. Yes glaucoma vs. No glaucoma					
	Medeiros et al. [53]	2019	1. RNFL thickness prediction	ResNet34	80%	83.7%		0.944
			2. Glaucoma vs. Suspect/healthy					
	Jammal et al. [52]	2020	RNFL prediction	ResNet34				0.801
	Lee et al. [62]	2021	RFNL prediction	M2M				
	Medeiros et al. [54]	2021	Detection of RFNL thinning from fundus photos	CNN				
**OCT 2D**	Asaoka et al. [55]	2019	Early POAG vs. no POAG	Novel CNN	80%	83.3%		0.937
	Muhammad et al. [63]	2017	Early glaucoma vs. health/suspected eyes	CNN + transfer learning			93.1%	0.97
	Lee et al. [64]	2020	GON vs. No GON	CNN (NASNet)	94.7%	100%		0.990
	Devalla et al. [65]	2018	Glaucoma vs. normal	Digital stain of RNFL	92%	99%	94%	
	Wang et al. [56]	2020	Glaucoma vs. no glaucoma	CNN + transfer learning				0.979
	Thompson et al. [51]	2020	POAG vs. no glaucoma	ResNet34	95%	81%		0.96
			Pre-perimetric vs. no glaucoma		95%	70%		0.92
			Glaucoma with any VF loss (perimetric) vs. no glaucoma		95%	80%		0.97
			Mild VF loss vs. no glaucoma		95%	85%		0.92
			Moderate VF loss vs. no glaucoma		95%	93%		0.99
			Severe VF loss vs. no glaucoma		95%	98%		0.99
	Mariottoni et al. [66]	2020	Global RNFL thickness value	ResNet34				
**OCT 3D**	Ran et al. [58]	2019	Yes GON vs. No GON	CNN (NASNet)	89%	96%	91%	0.969
					78–90%	86%	86%	0.893
	Maetschke et al. [57]	2019	POAG vs. no POAG	Feature-agnostic CNN				0.94
								0.92
	Russakoff et al. [59]	2020	Referable glaucoma vs. non-referable glaucoma	gNet3D-CNN				0.88
**AS-OCT**	Fu et al. [60]	2019	Open angle vs. Angle closure	VGG-16 + transfer learning	90%	92%		0.96
	Fu et al. [67]	2019	Open angle vs. Angle closure	CNN				0.9619
	Xu et al. [61]	2019	1. Open angle vs. angle closure	CNN (ResNet18) + transfer learning				0.928
			2. Yes/PACD vs. no PACD					0.964
	Hao et al. [68]	2019	Open angle vs. Narrowed Angle vs. Angle closure	MSRCNN				0.914

ARCH = Architecture; SEN = Sensibility; SPEC = Specificity; ACC = Accuracy; AUC = Area under the curve.

**Table 3 ijms-24-02814-t003:** Summary of studies on artificial intelligence applied to visual field testing.

Author	Year	Outcomes Mesures	Architecture	SEN	SPEC	ACC	AUC
Asaoka et al. [69]	2016	Pre-perimetric VFs vs. VFs in healthy eyes	FNN				0.926
Kucur et al. [75]	2018	Early glaucomatous VF loss vs. no glaucoma	CNN with Voronoi representation				
Li et al. [12]	2018	Glaucomatous VF loss vs. no glaucoma	VGG I 5	93%	83%	88%	0.966
Li et al. [76]	2018	Glaucoma vs. Healthy	VGG	93%	3%		0.966
Berchuck et al. [77]	2019	Rates of VF progression compared to SAP MD; Prediction of future VF compared to point-wise regression predictions	Deep variational autoencoder				
Wen et al. [78]	2019	HFA points and Mean Deviation	CascadeNet-5				
Kazemian et al. [74]	2018	Forecasting visual field progression	Kalman Filtering Forecasting				
Garcia et al. [73]	2019	Forecasting visual field progression	Kalman Filtering Forecasting				
DeRoos et al. [72]	2021	Forecasting visual field progression	Kalman Filtering Forecasting				

SEN = Sensibility, SPEC = Specificity, ACC = Accuracy, AUC = Area under the curve.

**Table 4 ijms-24-02814-t004:** Summary of studies on proteins and hormones used as biomarkers for glaucoma.

Author	Year	Type of Glaucoma	Biomarker	Type	Mechanism	Sample
Farkas et al. [81]	2004	Glaucoma	Transferrin, ceruloplasmin, ferritin	Peptides and Amino Acids	Upregulation	Serum
Lin et al. [82]	2014	Glaucoma	Ferritin	Peptides and Amino Acids	Upregulation	Serum
Wang et al. [83]	2018	Acute primary angle closure (APAC), non-glaucomatous cataract	Main matricellular proteins (SPARC, thrombospondin-2, and osteopontin)	Peptides and Amino Acids	Upregulation	Aqueous humor
González-Iglesias et al. [84]	2014	POAG, PEXG	Panel of 17 most differentially altered proteins	Peptides and Amino Acids		Serum
Ghaffariyeh et al. [95]	2009	POAG	Brain-derived neurotrophic factor (BDNF)	Peptides and Amino Acids	Downregulation	Serum
Ghaffariyeh et al. [95]	2011	NTG	Brain-derived neurotrophic factor (BDNF)	Peptides and Amino Acids	Downregulation	Tear
Oddone et al. [85]	2017	POAG	Brain-derived neurotrophic factor (BDNF)	Peptides and Amino Acids	Downregulation	Serum
Oddone et al. [85]	2017	POAG	Nerve growth factor (NGF)	Peptides and Amino Acids	Downregulation	Serum
Gupta et al. [88]	2017	POAG	Proinflammatory cytokines (IFNγ, IL-10, IL-12p70, IL-13, IL-1β, IL-2, IL-4, IL-6, IL-8, and TNFα)	Peptides and Amino Acids		Human tear samples
Guo et al. [89]	2019	POAG, CACG, PACS, AACG	TGFβ2, SFRP1	Peptides and Amino Acids		Aqueous humor
Li et al. [90]	2020	PACG	17-β-estradiol (E2), interleukin-8 (IL-8)	Peptides and Amino Acids		Serum
Canizales et al. [93]	2016	POAG	Superoxide dismutase 1 (SOD1)	Hormones and enzymes		Peripheral blood
Mirzaei et al. [96]	2020	Glaucoma	Complement pathway	Peptides and Amino Acids	Regulation	Peripheral blood

**Table 5 ijms-24-02814-t005:** Summary of studies on metabolic biomarkers and glaucoma.

Author	Year	Type of Glaucoma	Biomarker	Type	Mechanism	Sample
Lee et al. [96]	2017	Glaucomatous RNFL Defect	Homocysteine (Hcy)	Peptides, Amino Acids	Upregulation	Plasma
Lin et al. [97]	2020	POAG, NTG, OHT	Homocysteine (Hcy)	Peptides, Amino Acids	Upregulation	Plasma
Lin et al. [97]	2020	POAG, NTG, OHT	L-cysteine (Cys)	Peptides, Amino Acids	Upregulation	Plasma
López-Riquelme et al. [98]	2015	POAG, NTG	Homocysteine (Hcy)	Peptides, Amino Acids	Upregulation	Serum
López-Riquelme et al. [98]	2015	POAG, NTG	Endothelin-1 (ET-1)	Peptides, Amino Acids	Upregulation	Serum
Leibovitzh et al. [99]	2016	Glaucoma	Homocysteine (Hcy)	Peptides, Amino Acids	Upregulation	Plasma
Baumane et al. [100]	2017	Glaucoma and cataract	N-terminal fragment of the proatrial natriuretic peptide (NT-proANP, 1–98)	Peptides, Amino Acids	Upregulation	Plasma and aqueous humor
Javadiyan et al. [101]	2012	Glaucoma	Asymmetric dimethylarginine (ADMA), a dimethylated isomeric derivative of the amino acid l-arginine	Peptides, Amino Acids	Upregulation	Serum
Javadiyan et al. [101]	2012	Glaucoma	Symmetric dimethylarginine (SDMA), a dimethylated isomeric derivative of the amino acid l-arginine	Peptides, Amino Acids	Upregulation	Serum
Li et al. [104]	2019	POAG	Uric acid			Serum
Golubnitschaja et al. [105]	2007	Glaucoma	Stress response, apoptosis, DNA repair, cell adhesion, tissue remodeling, transcription regulation, multi-drug resistance, and energy metabolism	Peptides, Amino Acids		Circulating leukocytes in serum
Kotikoski et al. [106]	2002	Glaucoma	NOx (nitrite + nitrate), nitrite and cGMP	Peptides, Amino Acids		Serum and aqueous humor
Baudouin et al. [107]	1994	POAG	Inflammatory antigens	Peptides, Amino Acids	Upregulation	Conjunctival antigens

**Table 6 ijms-24-02814-t006:** Summary of studies on antibodies used as biomarkers in glaucoma.

Author	Year	Type of Glaucoma	Biomarker	Type	Sample
Hohenstein-Blaul et al. [109]	2017	POAG	Anti-GFAP, anti-γ-synuclein, and anti-myoglobin antibody as a control	Autoantibodies and Antibodies	Aqueous humor and tears
Beutgen et al. [108]	2019	POAG	Antibodies against trabecular meshwork	Autoantibodies and Antibodies	Serum
Grus et al. [110]	2006	POAG, NTG	IgG autoantibody, α-fodrin	Autoantibodies and Antibodies	Serum
Joachim et al. [111]	2005	POAG, NTG	IgG	Autoantibodies and Antibodies	Serum
Joachim et al. [112]	2007	POAG, PEX	IgG (heat shock protein 27, α-enolase, actin, and GAPDH)	Autoantibodies and Antibodies	Aqueous humor
Tezel et al. [113]	2012	POAG	IgG	Autoantibodies and Antibodies	Serum
Schmelter et al. [114]	2017	POAG	Autoantibody, IgG	Autoantibodies and Antibodies	Serum

GFAP = Glial fibrillary acidic protein, GAPDH = Glyceraldehyde 3-phosphate dehydrogenase.

**Table 7 ijms-24-02814-t007:** Summary of studies on possible applications of exosomes in glaucoma.

Author	Year	Materials	Target	Results
Lerner et al. [116]	2017	Cultured non-pigmented ciliary epithelium (NPCE) cells	Wnt signaling protein expression in the TM cells	>2-fold decrease in the level of β-catenin in the cytosolic fraction
Lerner et al. [117]	2020	NPCE primary cells	Wnt proteins in a human primary trabecular meshwork (TM) cells	Diminished pGSK3β phosphorylation and decreased cytosolic levels of β-catenin in primary TM cells. At the molecular level, it downregulated the expression of positive GSKβ regulator-AKT protein but increased the levels of GSKβ negative regulator-PP2A protein in TM cells.
Stamer et al. [118]	2011	Primary cultures of human TM cell monolayers		TM exosomes have a characteristic exosome protein profile and contain unique proteins, including the glaucoma-causing protein, myocilin.
Mead et al. [119]	2018	Bone marrow-derived stem cell (BMSC) small extracellular vesicles (sEV) and control fibroblast-derived sEV were intravitreally injected into 3-month-old DBA/2J mice once a month for 9 months.	Retinal ganglion cell (RGC) neuroprotection promoted by BMSC sEV	DBA/2J mice developed chronic ocular hypertension beginning at 6 months. The delivery of BMSC sEV, but not fibroblast sEV, provided significant neuroprotective effects for RBPMSþ RGC while significantly reducing the number of degenerating axons seen in the optic nerve. BMSC sEV significantly preserved RGC function in 6-month-old mice but provided no benefit at 9 and 12 months. BMSC sEV are an effective neuroprotective treatment in a chronic model of ocular hypertension
Pan et al. [120]	2019	Umbilical mesenchymal stem cells derived exosomes (UMSC-Exos) in a rat optic nerve crush (ONC) model		UMSC-Exos significantly promoted Brn3a+ RGCs survival in the retinal ganglion cell layer compared with PBS controls. UMSC-Exos also significantly promoted GFAP+ glia cell activation in retina and optic nerve. However, no increase in GAP43+ axon counts in the optic nerve was found after UMSC-Exos treatment.

**Table 8 ijms-24-02814-t008:** Summary of studies on genetics and precision medicine application to glaucoma management.

Author	Year	Type of Glaucoma	Gene	Chromosome	Location	Function
Morisette et al. [125]	1995	Primary Open Angle Glaucoma	GLC1A	1q23-q25	DlS445, DlS416/D1S480	Glycoprotein with a myosin-like domain, a leucine zipper region and an olfactomedin domain
Tamm et al. [124]	2002	Primary Open Angle Glaucoma	MYOC	1q24.3	Q368X	Glycoprotein with a myosin-like domain, a leucine zipper region and an olfactomedin domain
Scheetz et al. [126]	2016	Primary Open Angle Glaucoma	TMCO1	1q24.1		Transmembrane protein
Trikha et al. [127]	2015	Primary Open Angle Glaucoma	TGFBR3-CDC7	1p22.1	rs1192415	Cell-surface chondroitin sulfate/heparan sulfate proteoglycan
McCarty et al. [129]	2008	Primary Open Angle Glaucoma				
Sakurai et al. [130]	2014	Primary Open Angle Glaucoma, Normal Tension Glaucoma, Ocular Hypertension	Prostaglandin F2α (FP) receptor	1p31.1	rs12093097	Prostaglandine receptor
Low et al. [131]	2020	Malignant Glaucoma	BEST1	11q12.3	c.602 T > C, c.454 C > G, c.481 + 1 G > T, c.914 T > C	Transmembrane protein
Fini et al. [133]	2017	Steroid-induced Glaucoma	GPR158, HCG22	10p12.1 (GRP158), 6p21.33 (HCG22)		Cell-surface protein with seven transmembrane (7TM) domain
Patel et al. [134]	2013	Steroid-induced Glaucoma	GRP158	10p12.1		Cell-surface protein with seven transmembrane (7TM) domain
Jeong et al. [132]	2015	Steroid-induced Glaucoma	HCG22	6p21.33		
Yu-Wai-Man et al. [135]	2017	Primary Open Angle Glaucoma	IL6, PRG4	7p15.3 (IL6); 1q31.1 (PRG4)		Cytokine and proteoglycan
Zimmermann et al. [136]	2013	Primary Open Angle Glaucoma	IL-6 (IL-6–174G > C)	7p21	rs1800795	Proinflammatory cytokine
Lin et al. [137]	2014	Normal Tension Glaucoma	IL-6 (IL-6–174G > C)	7p21	rs1800795	Proinflammatory cytokine
Zhou and Liu [141]	2010	Primary Open Angle Glaucoma	IL-6	7p21	rs1524107	Proinflammatory cytokine
Fernando et al. [138]	2018	Primary Open Angle Glaucoma	siRNA nanocomplexes			RNAi induced by double-stranded small interfering RNA
McDonnel et al. [140]	2016	Primary human normal (NTM) with glaucomatous (GTM) cells; NTM cells under hypoxic conditions	TGFβ1, RASAL1	19q13.2 (TGFβ1), 12q24.13 (RASAL1)		Profibrotic factors

**Table 9 ijms-24-02814-t009:** Summary of studies on redox state and glaucoma.

Author	Year	Disease	Biomarker	Type	Mechanism	Sample
Petriti et al. [142]	2021	POAG	NAD+/NADH redox state	Vitamin	Upregulation	Lymphocytes from blood
Kouassi Nzoughet et al. [143]	2019	POAG	Nicotinamide	Vitamin	Downregulation	Plasma
Williams et al. [144]	2017	POAG	Nicotinamide Adenine Dinucleotide	Vitamin	Downregulation	Plasma
Hui et al. [145]	2020	POAG	Nicotinamide	Vitamin	Downregulation	Plasma
Salech et al. [146]	2020	Alzheimer’s Disease	Nicotinamide	Vitamin	Downregulation	Cerebrospinal fluid, microglia
Avery et al. [147]	2009	Slow Wallerian degeneration	Nicotinamide mononucleotide adenylyltransferase	Enzyme	Upregulation	Axons
Braidly et al. [148]	2009	Brain diseases	Inducible NOS, neuronal NOS	Enzyme	Upregulation	Human brain cells
Neufeld et al. [149]	1997	POAG	NOS-1, NOS-2, NOS-3	Enzyme	Upregulation	Optics nerve head
Balayage et al. [150]	2012	Glaucoma and optic neuropathy	SIRT1	Histone deacetylase	Uperegulation	Retinal ganglion cell

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
