# Peer review of "Precision Medicine in Glaucoma: Artificial Intelligence, Biomarkers, Genetics and Redox State"

_ijms, 2023, doi:10.3390/ijms24032814_

Round 1

Reviewer 1 Report

The authors made a comprehensive review on Precision Medicine in Glaucoma, which is interesting and can provide guidance to our clinical work. However, several issues should be addressed before possible publication:

1.       What’s the methods of literature search and what are the inclusion criteria of the literatures? It seems the literatures are not inclusive. I will take ‘UA’ for an example, the recently published review (PMID: 36088997) can help the authors make a more objective and comprehensive discussion on UA. Please check the references of each section.

2.       I suggest the authors make a brief discussion (conclusion, limitation or explanation of discrepancies between studies, etc.) at the end of section instead of just laying out the results of the references.

3.       The tables should be re-organized. In table 1, all the ‘sources’ are the same and can be deleted. In table 4, the type of glaucoma was described in the column of ‘No of patients’, ‘case control ratio’ was not calculated in all the tables, ‘n’ can be deleted in the column of ‘No of patients’. Please revise all the tables to make them more concise and readable.

4.       The keywords are missing.

5.       There are too many paragraphs in section ‘Introduction’.

Reviewer 2 Report

In this review, Maria Fea et al., tried to summary the state of art of precision medicine focusing in glaucoma diagnosis. The content is interesting, and it gives a good overview, however the presentation is somewhat bleak

-       In the body of most sections, studies are listed rather than discussed, and there are few occasions when the authors comment on the broad conclusions made by these studies and what the consensus among them is

-       The formatting of the tables should be re-organized for better readability; words are often cut in half in new lines, the headers, do not follow onto different pages, and the tables are very big

-       A reference list at the end with some of the more specific acronyms used would be helpful

Introduction:

-       I would have liked to see a more in-depth description of the biological and cellular bases of glaucomas at different stages rather than the unnecessarily long definition of biomarkers

-       Line 26: why are the early stages of glaucoma hard to detect currently? 

Artificial intelligence in glaucoma:

-       A lot of great references are listed to learn about the topic of AI and its use to diagnose glaucomas, but I would have liked to see more of a discussion on the overall state of the field rather than a presentation that reads more like a list of references

-       A figure visualizing the types of images or data that the different AIs are trained on would be helpful

-       For those of us not familiar with the differences in different types of AI networks used in this field (eg. CNN vs ResNet vs Inception), I would have liked to see a brief sentence stating their differences and tradeoffs

-       Line 68/69: A source or further discussion would be helpful

Biomarkers and precision medicine in glaucoma:

-       I thought this section was much stronger but still reads more like a list rather than an evaluation of the current state of the field

-       It would have been interesting to read which of the biomarkers they discuss would be more or less feasibly helpful in a clinical setting and why, as well as which have been found to have more predictive power in glaucoma detection

Genetics and precision medicine in glaucoma:

-       Lines 390-405: Is the mechanism for these SNPs leading to higher risk of POAG understood?

-       Lines 439: Could expand on the current limitations to the application of these findings in a clinical setting

NAD+/NADH redox state and glaucoma:

-       I thought that the introduction to this section was the best description so far and would suggest that they model the rest of their sections after this one 

-       Line 502/503: In which settings has this been tested so far?

Round 2

Reviewer 1 Report

The manuscript is more readable and comprehensive after the authors' revision. I have no other comments.

Reviewer 2 Report

The authors addressed the main points and provided a new version with major reviews.